# The Indian summer monsoon climate during the Last Millennium, as simulated by the PMIP3

Charan Teja Tejavath<sup>a</sup>, Karumuri Ashok<sup>a</sup>, Supriyo Chakraborty<sup>b</sup> and Rengaswamy Ramesh<sup>c</sup> Corresponding author: ashokkarumuri@uohyd.ac.in

<sup>a</sup>University centre for Earth and Space Sciences, University of Hyderabad, Hyderabad, India. <sup>b</sup>Indian Institute of Science and Tropical Meteorology, Pune, India. <sup>c</sup>School of Earth and Planetary Sciences, NISER, Bhubaneswar, India

10

5

## Abstract

Here, using the available model simulations from the PMIP3, we study the mean summer (June-September; JJAS) climate and its variability in India during the Last Millennium (CE 850-1849; LM) for

- which instrumental observations are unavailable, with emphasis on the Medieval Warm Period (MWP; CE 1000-1199 as against the CE 950-AD1350 from the proxy-observations) and Little Ice Age (LIA; CE 1550-1749 as against the CE 1500-1850 from proxy observations). Out of the eight available models, by validating the corresponding simulated global and Indian mean summer temperatures and mean Indian summer monsoon rainfall (ISMR), and their respective trends, from historical simulations (CMIP5) against
- the various observed/reanalysed datasets for the 1901-2005 period. From this exercise, we identify seven 'realistic' models.

The models simulate higher (lower) mean summer temperatures in India as well as globally during the MWP (LIA) as compared to the corresponding LM statistics, in confirmation of several proxy data sets. Our Analysis shows a strong negative correlation between the NINO3.4 index and the ISMR and a positive

Our Analysis shows a strong negative correlation between the NINO3.4 index and the ISMR and a positive correlation between NINO3.4 and summer temperature over India during the LM, as is observed in the last one-and-half centuries. The magnitude of the simulated ISMR-NINO3.4 index correlations, as seen from the multi-model mean, is found to be higher for the MWP (-0.19; significant at 0.05 level) as compared to that for the LIA (-0.09; insignificant). Our analysis also shows that the above (below) LM-mean summer

temperatures during the MWP (LIA) are associated with relatively higher (lower) number of concurrent El Niños as compared to the La Niñas. Distribution of boreal summer velocity potential at 850 hPa in the central tropical pacific and a zone of anomalous convergence in the central tropical pacific, flanked by two zones of divergence in the equatorial pacific, suggesting a westward shift in Walker circulation as compared to the current day signal. The anomalous divergence centre in the west also extends into the

equatorial eastern Indian Ocean, which results in an anomalous convergence zone over India and therefore excess rainfall during the MWP as compared to the LM. The results are qualitative, given the inter-model spread.

## 5 Introduction

Instrumental records of climate seldom date back prior to the 1850s. Therefore, analysis of proxy climate data, aided by climate modelling, has been the principal means to evaluate past climate variability. Past climate records exhibit significant variability on millennial to inter-annual time scales (IPCC, 2013).

- Interestingly, this IPCC report based on a review of several publications, points out significant centennial climate variations during the last two millennia, though there is no apparently significant change in the external climate forcing from the first half of 20th century. Paleo-data based studies such as those by Lamb, 1965; Grove, 1988; Graham et al., 2010 identify two significant events in the last millennium (LM) prior to the period when instrumental observations stated, i.e. Common Era (CE) 850-1849. These are, (i) a relatively warmer period known in literature as the 'Medieval Warm Period' (MWP, CE 950-1350), roughly
- followed by (ii) a relatively cooler period, the Little Ice Age (LIA, CE 1500 1850). Presence of these warmer (MWP) and cooler (LIA) periods varies from region to region, in terms of timing, duration and magnitude of the temperature anomalies.
- 20 Paleoclimate reconstructions from various well-dated proxy data suggest that during the MWP, some regions experienced temperatures as warm as mid-20<sup>th</sup> century whereas some others were as warm as the late-20<sup>th</sup> century (e.g., IPCC 2013, Prasad and Enzel, 2006; Fleitmann et al., 2007; Borgaonkar et al., 2010; Ponton et al., 2012).
- 25 The Indian Summer Monsoon Rainfall (ISMR; June-September; JJAS) variability is manifested on intraannual, inter-annual, decadal, centennial and millennial to multi-millennial time scales (Ramesh et al., 2011). Paleo-monsoon records from well-dated proxy data show centennial-to millennial-scale changes in the ISMR during the Holocene: from the Arabian Sea (e.g. Sarkar et al., 2000; Gupta et al., 2003; Staubwasser et al., 2003; Tiwari et al., 2005), the Arabian Peninsula (e.g. Fleitmann et al., 2007; Fleitmann et al., 2003;Neff et al., 2001), and the Indian sub- continent (e.g. Berkelhammer et al., 2012; Dixit et al., 2013;
- 2015; Dixit et al., 2014a; Dixit et al., 2014b; Dixit, 2013; Dutt et al., 2015; Nakamura et al., 2015).

In a recent review, Dixit and Tandon (2016) suggest that MWP and LIA effects are well reflected in the ISMR, with a caveat that proxy data exhibit heterogeneity in terms of the timing and duration. Proxy

records also suggest that, by and large, during the last millennium, ISMR was the highest during the MWP and relatively weaker during the LIA (Yadava et al., 2005). However, the data density is rather sparse in time and space to quantify the decadal scale temporal structure of ISMR variability during MWP and LIA.

- 5 A speleothem-based reconstruction of ISMR variability by Sinha et al., (2007) exhibits an evolution conforming to solar activity (a surrogate for which is radiocarbon activity) only during the MWP. An increased summer monsoon precipitation during the MWP is suggested to be linked to the ENSOmodulated solar forcing in proxy studies by Berkelhammer et al., (2010) and Emile-Geay et al., (2007). The speleothem-based monsoon reconstruction of Sinha et al., (2007 and 2011) suggests a severe 10 weakening of ISM (Indian Summer Monsoon) during the LIA, apparently associated with (multi-year to
- decades long duration of) droughts particularly between 13<sup>th</sup> and 17<sup>th</sup> centuries. Another proxy record, from the Dandak cave in Central India, shows a 30% rainfall reduction during the 14<sup>th</sup> century (Yadava, et al., 2005).
- 15 Obviously, the recent ~150-year period is the best documented period in terms of instrumental observations. Uncertainties, however, exist in terms of the quality and spatial density of data even for this period.
- The observational records of ISMR from the beginning of last century show that its inter-annual and inter-decadal variability is significantly associated with that of the El Niño-Southern Oscillation (ENSO). Typically, the warmer (cooler) ENSO events are associated with lesser (higher) than normal rain over India during the boreal summer, concurrent with the Indian monsoon season. Prasad et al. (2014) infer that the long-term influence of ENSO like conditions on ISM began only 2ky BP, and is coincident with Southern Indo-Pacific warm pool (IPWP) warming, based on proxy climate data. They also suggest that the IPWP-ISM links and large scale advection of moist air toward India varies on a multi-centennial scales. Kitoh et al. (2007), in a model study, observed decadal variability in the ENSO-ISM relation. Through a 31-yr moving correlation analysis, they show that, during the LM, monsoon-ENSO correlations vary over a wide range, specifically -0.71 to +0.07, with an overall correlation of -0.34 for the LM.
- 30 Thus, the variability of Indian summer monsoon during the LM has been relatively less studied, particularly from the modelling perspective. It is also noticeable that all the model studies cited above primarily employed *single* GCMs. From this perspective, it is interesting to explore multi-model simulations such as those from the PMIP3, to study Indian summer monsoon conditions during the LM, specifically the MWP and the LIA, and examine whether these model results could be reconciled with

proxy-observations. Likewise, such a study highlights the capability of these models in capturing at least a millennium of the past climate with fidelity, in addition to facilitating a quantification of the multi-model spread. Furthermore, such a study would serve as a benchmark for addressing longer periods of climate variability relevant to the Indian summer monsoon using models.

5

With this motivation, here we study the multi-model simulated ISMR variability and its teleconnections with the ENSO during the LM, using various relevant PMIP3 datasets with an emphasis on the Medieval Warm Period (MWP; CE 1000-1199) and Little Ice Age (LIA; CE 1550-1749). We consider 200 warmest (relatively coldest) years from MWP (LIA) for maintaining uniformity between global and

10 regional analysis of ENSO-ISM teleconnections from CMIP5 LM simulations, with the knowledge that the signatures of the MWP and LIA varied from region to region, at least in terms of magnitude (e.g. Dixit and Tandon 2016).

In the following sections, we describe the various reanalysed, observed, and PMIP3 datasets we used, present our results subsequently, and finally provide a concluding summary.

# Data and Methodology

- It is indeed a challenging prospect to validate the PMIP3 simulations for the LM period over India 20 given the sparse and scanty observations. Fortunately, model simulations of the CMIP5 for the historical 20 period (CE 1850-2005) can be validated using various observed/reanalysed gridded datasets, keeping in 20 mind the uncertainties associated with such datasets during the pre-satellite period. Therefore, in this study, 20 we start by exploring the fidelity of simulated Indian summer monsoon climate from historical simulations 20 (henceforth referred to as HS) that cover the CE 1850-2005 period for which instrumental observations are
- 25 available. It may be noted that this exercise is carried out only for eight CMIP5 models for which the PMIP3 simulations for the LM period are available for the CE 850-1849 period (LM), under the class termed as 'past1000 (henceforth referred to as p1000)'.
- From the HS, the models were forced using the observed atmospheric composition changes with all natural aerosols or their precursors, and natural sources of short-lived species, and time-evolving land cover as outlined by Taylor et al. (2012). On the other hand, the p1000 results were obtained by forcing the models with well-mixed greenhouse gases, changes in volcanic aerosols, land use, and solar irradiance changes (Taylor et al, 2012). We evaluate the fidelity of the HS simulations by comparing with the observed/reanalysed Indian summer monsoon rainfall and air temperature. This helps us to identify the

'good' PMIP3 models for further analysis. The seven models whose data used in this study are: BCC-CSM-1-1(m), IPSL-CM5A-LR, FGOALS-s2, MIROC-ESM, MPI-ESM-P, GISS-E2-R, CCSM4 and HadCM3. These datasets have been downloaded from "<u>http://cera-www.dkrz.de/WDCC/ui/Index.jsp</u>". The acronyms used and details for these datasets are presented in Table-1; various observational/reanalysed data

- sets used for the validation of the HS are, the Hadley Centre Interpolated sea surface temperature (HadISST; Titchner and Rayner, 2014) for CE 1870-2014, the Extended Reconstructed Interim skin temperature, sea surface temperature and 2 m air temperature (ERA-Interim SKT, SST and T2M; Hersbach et al. 2015) available for CE 1900 to 2010. We also use the India Meteorological Department (IMD) gridded rainfall datasets for CE 1901-2009 period, available at 1.0° latitude x 1.0° longitude resolution
- and covering the land region bound by 66.5° E-101.5° E; 6.5° N-39.5° N (Rajeevan et al., 2006). For uniformity, all the simulated precipitation and near air surface temperature data sets were re-gridded to 2.0° latitude x 2.0° longitude resolution grids.

Table 1:- CMIP5/PMIP3 Last Millennium and Historical simulations, their acronyms and temporal coverage.

| S No | CMIP5/PMIP3    | p1000 (Last Millennium) simulation | Historical simulation temporal coverage | Acronyms |
|------|----------------|------------------------------------|-----------------------------------------|----------|
|      | Models         | temporal coverage                  |                                         |          |
| 1    | BCC-CSM-1-1(m) | CE 0850-1849                       | CE 1850 -2005                           | BCC      |
| 2    | CCSM4          | CE 0850-1849                       | CE 1850 -2005                           | CCSM4    |
| 3    | IPSL-CM5A-LR   | CE 0850-1849                       | CE 1850 -2005                           | IPSL     |
| 4    | MIROC-ESM      | CE 0850-1849                       | CE 1850 -2005                           | MIROC    |
| 5    | MPI-ESM-P      | CE 0850-1849                       | CE 1850 -2005                           | MPI      |
| 6    | GISS-E2-R      | CE 0850-1849                       | CE 1850 -2005                           | GISS     |
| 7    | FGOALS-s2      | CE 0850-1849                       | CE 1850 -2005                           | S2       |
| 8    | HadCM3         | CE 0850-1849                       | CE 1850 -2005                           | HADCM3   |

Given that model results exhibit significant variance, multi-model averaging is a well-accepted method to reduce the model biases and derive a consensus estimate. We therefore use the multi-model mean statistics as a part of the validation of the CMIP5 simulations. Unfortunately, from an analysis of the globally averaged temperature from the LM simulations, which will be discussed in more detail in *section 3.2*, it is apparent that the MIROC-ESM simulates a strong increasing trend in globally-averaged temperature (Figure 4), which is in disagreement with the evolution of the corresponding parameter by the other models. Nor is such a strong trend observable in proxy data (see Box TS.5, Figure-1 of Technical Summary in the IPCC 2013; Stocker 2013). Therefore, this model is not used in generating our multi-

model mean (MMM) statistics.

20

Thus, the historical simulations are validated by comparing various climate statistics from the MMM with the corresponding climate statistics from observed and reanalysed datasets for the CE 1901-

2005 period. Subsequently, we also discuss multi-model mean statistics with these models for the p1000 simulations.

We use the NINO3.4 index, an area-averaged SST over the region bound by 170°E-240°W; 5°S5°N. An The Indian summer monsoon rainfall (ISMR) index is obtained by area-averaging the mean June-through-September (JJAS) rainfall over the land region bound by 65° E-95° E; 10° N-30°N.

One must be aware that individual ENSO events may not occur at the same time in across all models, and consequently, the variance of the MMM ENSO index (such as Nino3.4 index) may decrease. 10 Indeed, as can be seen later, the standard deviation of the Nino3.4 index from the Multi-model mean (MMM) is 0.3 C as against the ~0.45 C for the index from individual models (Tables 4 and 7). Having said that, studies such as the Lewis & Legrande (2015), and Power et al., (2013) use MMM to demonstrate that the ENSO is still associated with the dominant EOF mode (e.g. Power et al., 2013) as well as its teleconnections (Lewis & Legrande 2015). The ENSO events simulated by the MMM are obviously very strong, and are robust to the "noise" from other concurrent model signals which could be neutral or even opposite-phased.

To check the ENSO-ISM relations during LM and to identify any significant interannual climate events during LM, we calculate the monthly anomalies of surface temperature and precipitation from their respective climatological monthly means. The anomalies of any parameter, such as, say, JJAS temperature, for each model have been obtained by subtracting the 1000-year climatological value from the individual seasonal values. Linear correlation analysis is used to estimate the ENSO-ISMR relationship during various periods.

## 25 3 Results

#### 3.1 Validation of the HS

Figure 1a and 1b respectively show a 11-year running mean of globally-averaged near surface air temperature from the eight models of the HS, along with the same from the MMM *constituted by the seven models* (as explained in the previous section); Figure 1c and 1d show the corresponding time series of anomalies (from the 11-year mean from the individual models as well as the MMM. It is seen from Figures 1c and 1d that all the models are able to simulate the observed increasing temperature trend reasonably, notwithstanding an inter-model spread. Further, we find that the observed as well as and the simulated trends are significantly above the corresponding interannual standard deviations (e.g. Figure

5

10

SPM.1a; Figure TS. 1; Figure TS. 9; Stocker et al., 2013; IPCC, 2013;). The Figure1d suggests that the surface temperatures over India also have continued to rise as we step into the 21<sup>st</sup> century, which is in agreement with observations (Revadekar et al., 2012).