# Peer review of "The Indian summer monsoon climate during the Last Millennium, as simulated by the PMIP3"

_Climate of the Past, 2017_

## Referee Comment (RC1) · O. Bothe (Referee) · 30 May 2017

Dear authors, and dear editor,

Regarding your manuscript titled "The Indian summer monsoon climate during the Last Millennium, as simulated by the PMIP3" (cp2017-24), I see a number of problems which you have to resolve. I'll detail these below. I am not quite sure whether I should recommend rejection or whether these concerns can be addressed in a substantial revision.

Let me first state, that a thorough analysis of the Indian Summer monsoon in the PMIP3-past1000-ensemble would be a valuable contribution to our understanding of

the climate of the past millennium.

Note to editor and authors: Please crosscheck the data mentioned on page 5 line 8 with the data-availability-requirements of CP.

**Main concerns:**

You present a set of simple correlation analyses, the frequency of ENSO-events for different periods, and the velocity potential over these periods for the PMIP3 multi-model ensemble. I am not sure whether this set of analyses is substantial enough to warrant publication.

Not least if I understand your analyses correctly - but you obviously can prove me wrong - parts of them are inappropriate or wrong which has implications for the results.

First, you diagnose differences in correlation coefficients between two periods. However, Gershunov et al. (2001, http://dx.doi.org/10.1175/1520-0442(2001)014<2486:LFMOTE>2.0.CO;2) and others discuss how correlation coefficients can change over time without a change in the underlying relation. My impression is, that maybe one if any correlation-difference may be significant in your analyses.

Secondly, if I understand your analyses of the ENSO-frequencies correctly, you do not identify differences in La Nina and El Nino frequencies but just diagnose differences in the mean background state.

This implies, I am sorry to say, If I am correct, that two of your three main results are incorrect.

**Further concerns:**

I am not sure your use of the Multi-Model-mean is appropriate. I think for most of your analyses the spread and the differences between the models are of interest and the multi-model-mean analysis is not necessary. For example correlating the MMM-ISMR- and MMM-NINO3.4-indices seems not very meaningful. As another example, I am not
sure, the premise of the paragraph on Page 6 Line 8ff is valid.

You often refer to proxies but don't show any comparisons at all. For example on page 12 line 3, it would help to be able to make the comparison directly or at least in an appendix.

P7L14 and L16: I do not see the decreasing precipitation trend. If it is there, please quantify it more clearly.

P12L7: I'd like to repeat, that I don't think these correlations mean much for the multi-model-ensemble-mean, as you don't identify the relation between ENSO and Indian climate by this, but just the common long term relation between the global mean and both the ENSO-region and India.
Additionally, the correlation coefficients are generally rather small, which to me also indicates that, at least for temperature, you just capture the concurrent relation between TG and TI and TG and NINO34.

Furthermore, you make some additional inferences which I would call wrong or at least your analyses do not warrant them, e.g.:

P11L25: "All this suggests a better agreement among the models in simulating the long term trend in rainfall over India relative to the variability during the MWP and that during the LIA."
I don't think your analyses allow to infer this.

P13L10ff: "we surmise that Indian Sub-continent was warmer (relatively cooler) during the CE 1000-1199 (CE 1550-1749) than the many other regions of the globe".
I don't understand how you come to this assumption, but I don't think you can infer this from comparing one regional series with the global mean.

P14L3: "More importantly, these results suggest that any long term weakening between the ENSO-Indian summer monsoon (e.g. Kumar et al., 1999; Ashok et al., 2001) is not necessarily due to anthropogenic climate change."

[Figure]

It's not that this isn't a possibility but from my point of view your data and your analyses do not give any indication of this, they in my opinion don't tell you anything at all about this.

P14L21: "It is known that the El Niños (La Niñas) cause anomalous increase (decrease) in global temperature. Therefore, a predominant presence of higher number of simulated El Niños as compared to La Niñas in almost all the models is the reason why the simulated MWP is warmer as compared to the LIA. Given this agreement across the models, we can surmise that, in real world too, the MWP is likely due to the occurrence of a relatively higher frequency of El Niños as compared to the La Niñas"
a) Can you give references for "It is known . . ."
b) You do not identify a change in frequency of El Ninos but just describe the temperature change in the background state.
c) That is, the MWP in the models isn't warmer and LIA isn't colder because of ENSO. Rather your analysis just describes the warmer MWP and the colder LIA.
d) Thus your inference is circular and in this context very likely wrong.

P19L2: "Our analysis of the PMIP3 datasets suggests that the Indian region was likely warmer than the global temperature during the MWP."
I don't think your analysis suggests this.

**Minor points:**

Is the title appropriate? I would say, you discuss more the ISM-ENSO relation than the general state of the ISM.

I would suggest to drop the MIROC-simulation completely from the paper. It is enough to mention from the start that you don't use it, because of the known problems with the simulation. There are a number of references available, I think.

I found the manuscript in parts hard to read, thus I would suggest to check where things could be rewritten to make the manuscript more clearly readable.

Page 1 Line 24: The models can't confirm the proxy-data, they can only be consistent with them or, if you insist, can agree with the proxies.

P2L11: What do you mean by "there is no apparently significant change in the external climate forcing from the first half of 20th century"?

P3L5: Is this full paragraph relevant for this paper?

P3L33: Please give a reference for PMIP3.

P4L19ff: I don't think you validate the models

P4L33: The correct references for the PMIP3-past1000 simulation-setup are Schmidt et al. (2011, doi:10.5194/gmd-4-33-2011) and Schmidt et al. (2012, doi:10.5194/gmd-5-185-2012)

P5L6: I don't think ERA-Interim goes from 1900 to 2010?
P5L8: Can you please provide a reference for the IMD-data? And is this data publically available? If yes, please provide a URL, if not please provide contact details where the data can be obtained. [Dear editor, please crosscheck this with the data-availability-requirements.]

P6L33ff: Is this relevant? What does this imply?

P7L11: You do not plot events but just the 11-running means which possibly masks the higher frequent ENSO-variability.

P8L18: You shortly write about standard deviations here and elsewhere. From my point of view, it does not become clear, what's the point of these discussions.

P10L16: Which models are these outliers, what is the bias because of which you call them outliers, where do you show this?

P11L2: Much of what you describe here for the global temperature is in the IPCC and other publications.

P11L18: I do not see these decreasing ISMR-trends, please clarify.

P11L20ff: I cannot really follow the premise of this sentence.
P11L22: Why would we "expect" this?

P12L11: I think you mean "1000-year".

P13L1ff: Maybe I miss something why are these statistics of interest/relevant?

P13L17: What do you mean by "realistic"?

P13L19: GOALS should read FGOALS.

Your Figure 7 does not show the ensemble but only one model, as far as I can see.

P15L4: What do you mean by this "discrepancy".
P15L5: What kind of "factor" is this "discrepancy" meant to be?

P17L7: Maybe you should discuss the different models first before writing about the composite.

P19L29: Can you provide a reference for this cautionary note.

P20L8: Maybe I missed it in your results-section but I think you should discuss these contrasts between the LM-relations and the modern relations in more detail and possibly show them in Figures or at least supplementary materials.

The Figures produced with GrADS are sometimes of suboptimal quality. Furthermore I recommend to change the color scale which is rather easy in GrADS if I recall correctly. The reason for this is, e.g., https://www.climate-lab-book.ac.uk/2014/end-of-the-rainbow/.

It is often unclear whether MMM refers to the multi-model-mean or to the multi-model-ensemble, its members, or its spread. One example is on page 8 in line 11.

Please do not insert tables as pictures into the manuscript.

You quite often put words or sentences in italics to emphasize them. I think this is unnecessary.

Please be sure that you refer in the text to the correct figures and tables.

I think you have to cite Rehfeld and Laepple (https://doi.org/10.1016/j.epsl.2015.12.020) in your paper.

I won't detail typos etc., as I think the CP-copy-editor is going to deal with them - but there are a number of them.

In conclusion, I recommend you concentrate on the dynamical differences between a/the warmer period/s and a/the colder period/s of the last millennium and their implications for the Indian Summer Monsoon, as this would fill a gap in our studies on the last millennium in simulations.

Best regards

―――――――――――――――――――

---

## Short Comment (SC1) · 3 Jun 2017

The PAGES Data Stewardship Integrative Activity seeks to advance best practices for sharing data generated and assembled as part of all PAGES-related activities. As part of this activity, a team of reviewers has been constituted for the "Climate of the Past 2000 years" Special Issue. The data team is reviewing the data handling within each of the CP-Discussion papers in relation to the CP data policy and current best practices. The team has identified essential and recommended additions for each paper, with the goal of achieving a high and consistent level of data stewardship across the 2k Special Issue. We recognize that an additional effort will likely be required to

meet the high level of data stewardship envisaged, and we appreciate the dedication and contribution of the authors. This includes the use of Data Citations (see example in supplement). We ask authors to respond to our comments as part of the regular open interactive discussion. If you have any questions about PAGES Data Stewardship principles, please contact any of us directly.

Best wishes for the success of your paper,

2k Special Issue Data Review Team (Darrell Kaufman, Nerilie Abram, Belen Martrat, Raphael Neukom, Scott St. George) and ex-officio team members (Marie-France Loutre, Lucien von Gunten)

For this paper:

Expand the "Data Availability" section to include a Data Citation or URL link to the primary results generated by this study. We request that the data displayed in the following figures be deposited in a public repository:

- Fig 1: modeled and instrumental air temperatures (past 100 years) for India and globally (Fig 1a and b)

- Fig 2: modeled and instrumental ISMR and NINO3.4 index (past 100 years) for India and globally (Fig. 2a, c and d)

- Fig 4: modeled air temperature (past 1000 years) for India and globally (Fig 4a and 4b)

- Fig 5: modeled ISMR (past 1000 years) for India and globally

- Fig 7: NINO 3.4 index from CCSM4 during MWP and LIA (strength and year)

- Fig 8: anomaly fields for rainfall, wind, and velocity potential for MWP and LIA from CCSM4

- Fig 9: anomaly fields for temperature and rainfall for MWP and LIA

Please also note the supplement to this comment:
http://www.clim-past-discuss.net/cp-2017-24/cp-2017-24-SC1-supplement.pdf
* * *
[Figure]

**Supplement:**

Data Citations track the provenance of a dataset giving credit to the data generator; this is in addition to any references to publications where the data are described. Data Citations are used in the text (or tables) alongside and in the same way as publication citations. In the Reference list, they include: Creators, Title, Repository, Identifier, Submission Year. More information about Data Citations is here: <https://www.datacite.org/mission.html>

Here is an example of text and corresponding citations (using CP punctuation style):

The PAGES2k Consortium (in press) assembled a large global dataset of temperature-sensitive proxy records (PAGES2k Consortium, 2017). Among the records is the paleo-temperature reconstruction from Laguna Chepical (de Jong et al., 2016), which was described by de Jong et al. (2013).

de Jong, R., von Gunten, l., Maldonado, A., and Grosjean, M.: Late Holocene summer temperatures in the central Andes reconstructed from the sediments of high-elevation Laguna Chepical, Chile (32° S), Climate of the Past, 9, 1921-1932, 2013.

de Jong, R., von Gunten, l., Maldonado, A., and Grosjean, M.: Laguna Chepical summer temperature reconstruction, World Data Center for Paleoclimatology, https://www.ncdc.noaa.gov/paleo/study/20366, 2016.

PAGES 2k Consortium: A global multiproxy database for temperature reconstructions of the Common Era, Scientific Data, in press.

PAGES 2k Consortium: A global multiproxy database for temperature reconstructions of the Common Era, version 2.0.0, figshare, https://figshare.com/s/d327a0367bb908a4c4f2, 2017.

---

## Short Comment (SC2) · 15 Jun 2017

How do you define the NINO3.4 index? I have seen in the manuscript, you have mentioned NINO3.4 index as an area-averaged SST over the region bound by 170°E-240°W; 5°S-5°N (Page 6, Line 4-5). However, the standard NINO3.4 region is bounded by 5°N to 5°S, from 120°W to 170°W. Please explain about this differences. Also, you may need to mention SST anomaly instead of SST (Page 6, Line 4).

---

## Referee Comment (RC2) · W. Man (Referee) · 25 Jul 2017

General comments: By using the available model simulations from the PMIP3, the authors study the mean summer climate and its variability in India during the Last Millennium, with emphasis on the Medieval Warm Period (MWP; CE 1000-1199) and Little Ice Age (LIA; CE 1550-1749). The models simulate higher (lower) mean summer temperatures in India as well as globally during the MWP (LIA) as compared to the corresponding LM statistics. The Analysis shows a strong negative correlation between the NINO3.4 index and the ISMR and a positive correlation between NINO3.4 and summer temperature over India during the LM, as is observed in the last one-andhalf centuries. The above (below) LM-mean summer temperatures during the MWP (LIA) are associated with relatively higher (lower) number of concurrent El Niños as compared to the La Niñas. There is a westward shift in Walker circulation during the MWP, and the anomalous divergence center in the west also extends into the equatorial eastern Indian Ocean, which results in an anomalous convergence zone over India and therefore excess rainfall. It is important to understand the Indian summer monsoon rainfall (ISMR) variance and aspects of its possible causes, simply because many severe social and economic impacts are associated with ISMR anomalies. This simulation result well describes major aspects of the ISMR and the possible dynamics. I regard the manuscript by Tejavath et al. is suitable to be published by Clim. Past after some moderate revisions. Specific Comments: (1) Paleoclimate reconstructions from proxy data suggest that during the MWP, a cooler tropical eastern Pacific, referred to as a La Niña-like background state, is reconstructed. However, this is not evident in the PMIP3 model simulations. Almost all the models except one consistently simulate more El Niños as compared to La Niñas during the MWP compared to the LIA. In P.14, the authors state that "It is known that the El Niños (La Niñas) cause anomalous increase (decrease) in global temperature. Therefore, a predominant presence of higher number of simulated El Niños as compared to La Niñas in almost all the models is the reason why the simulated MWP is warmer as compared to the LIA. Given this agreement across the models, we can surmise that, in real world too, the MWP is likely due to the occurrence of a relatively higher frequency of El Niños as compared to the La Niñas". I think this statement is not reasonable. The reconstruction exhibits a La Niña-like pattern in the tropical Pacific during the MCA (Cobb et al. 2003; Graham et al. 2007; Mann et al. 2009). Besides, the La Niña-like condition is reproduced in simulations employing the simplified Zebiak-Cane model of the tropical Pacific coupled ocean-atmosphere system (Mann et al. 2005), which exhibits a stronger dynamical feedback than most global models. Thus, it is not correct to say that in real world, the MWP is likely due to the occurrence of a relatively higher frequency of El Niños as compared to the La Niñas" just from the perspective of model results. The global

temperature changes may have been driven by the effective radiative forcing during the past millennium. However, there is little evidence for globally synchronized MCA and LIA intervals, with the specific timing of these intervals varying regionally, which may have been dominated by the internal variability. (2) The simulated ISMR anomaly shows a weak decreasing trend throughout the LM. The authors also attributed the possible dynamics to the more number of El Niños during the MWP as compared to the LM. The distribution of summer velocity potential at 850 hPa suggests a westward shift in Walker circulation, and the anomalous divergence center in the west also extends into the equatorial eastern Indian Ocean, which results in an anomalous convergence zone over India and therefore excess rainfall during the MWP. It is good that the model results are inter-consistent by themselves. Proxy records also suggest that the ISMR was higher during the MWP and relatively weaker during the LIA (Yadava et al. 2005). A speleothem-based reconstruction of ISMR variability exhibits an increased summer monsoon precipitation during the MWP and a severe weakening of monsoon rainfall during the LIA, apparently associated with droughts particularly between 13th and 17th centuries. However, proxy reconstructions show opposite ENSO conditions as compared with the simulations during the MWP and LIA periods, how can we explain the ENSO-monsoon relationship and the possible dynamics from the reconstruction perspective? (3) Apart from the Walker circulation changes, does the land-sea thermal contrast change in the upper-troposphere also play an important role for the ISMR variability during the LM? If yes, can we further attributed to the external forcing drivers? Since the correlations between ENSO and the ISMR may differ on the multi-decadal-to-centennial scales from that on the inter-annual timescales. Typing errors: (1) P.16, Line 16, there are two "due to", delete one.

Please also note the supplement to this comment:
https://www.clim-past-discuss.net/cp-2017-24/cp-2017-24-RC2-supplement.pdf

---

## Author Comment (AC1) · 18 Aug 2017

At the outset, we would like to thank the reviewer for the useful and encouraging comments, which have improved the standard of the manuscript.

Main concerns:

Question 1: First, you diagnose differences in correlation coefficients between two periods. However, Gershunov et al. (2001, http://dx.doi.org/10.1175/1520-442(2001)014<2486:LFMOTE>2.0.CO;2) and others discuss how correlation coefficients can change over time without a change in the underlying relation. My impression

is, that maybe one if any correlation-difference may be significant in your analyses.

Response:

Thank you for the comment. Gershunov et al., 2001 paper cautions about the sampling issues associated with a short (21-31year) running correlation within a 130-150 yrs period, which does not vary in terms of external forcings (i.e. change in solar signal, or significant volcanic eruptions etc.). In such a case (specifically, weakening of ENSO-Indian summer monsoon rainfall in the last 15-20years of the 20th century), it is very important to check if any such change in decadal variability may be subject to stochasticity.

However, in our case, we evaluate changes in a statistic (correlation) between two separate climatic (200 year) regimes, with different external forcings such as different frequencies of volcanic eruption, etc. Therefore, our results are not subject to the Gershunov et al., 2001 concern.

As we have shown in the manuscript, the difference between the ENSO-ISMR correlation from the MWP simulations and the LIA simulations are statistically significant at 0.05 level from a Student's two tailed t-test.

Having said this, we have anyway carried out a bootstrapping significance test for the above mentioned difference in correlation (1000 simulations). We find that the results from 4 out of 7 models are significant at 95% confidence level.

Question 2: Secondly, if I understand your analyses of the ENSO-frequencies correctly, you do not identify differences in La Nina and El Nino frequencies but just diagnose differences in the mean background state.

Response:

Thank you. Actually, we had shown the difference in the frequencies of El Niños and La Niñas to explain the relatively warm MWP and relatively cold LIA temperatures. Specifically, the MWP simulations exhibit more number of El Niños relative to the La

Niñas. It is well known that the El Niños (La Niñas) result in warmer (cooler) than normal global temperatures (e.g. Trenberth et al., 2002). Therefore, the simulated above normal global temperatures during the MWP are in conformation with the relatively more number of simulated El Niños. The simulated relatively cooler LIA is also easily explained by the simulated ENSO frequency skewed negatively, that is more simulated La Niñas as compared to El Niños during this period. Our results also clearly indicate (Figure R1) that the simulated global mean temperature is positively and significantly correlated with the Nino3.4 index during the LM and its sub-periods.

The differences in the simulated mean background state shown, on the other hand, explain why the simulated ISMR during the MWP is more than that during the LIA. For example, the 'anomalies' in large scale convergence and divergence patterns during the MWP relative to the LM averages explain that there is a relatively eastward shift in Walker circulation, with associated changes over India, which have resulted in more rainfall during this period as compared to the LM. To be clear, the correlations of the simulated Nino3.4 index with local summer monsoon temperatures over various places in India (Figure-A4 of the revised manuscript, attached below as Figure R1), and that with the ISMR (Figure-A5, attached below as Figure R2) are all negative, as is observed in general during the last 150years or so.

We must mention that, in the revision, we have refined the frequency distinction analysis for the sake of a better objectivity, notwithstanding that the result is qualitatively the same as compared to that from the earlier version of the manuscript. Specifically, the earlier version, we had identified a simulated ENSO event by fixing an amplitude-threshold of 0.5°C for the Nino3.4 index. Now, we identify a simulated ENSO event when the amplitude of the NINO3.4 index exceeds its standard deviation (If temperature is below one standard deviation considered as La Nina and if it is above one standard deviation, it is considered as strong El Niño or La Niña events). We still find that there are relatively more El Niños (La Niñas) during simulated MWP (LIA) as compared to LIA (MWP).

[Figure]

Trenberth, K. E., J. M. Caron, D. P. Stepaniak, and S. Worley, Evolution of El Niño–Southern Oscillation and global atmospheric surface temperatures, J. Geophys. Res., 107(D8), doi:10.1029/2000JD000298, 2002.

Further concerns:

Concern:

I am not sure your use of the Multi-Model-mean is appropriate. I think for most of your analyses the spread and the differences between the models are of interest and the multi-model-mean analysis is not necessary. For example correlating the MMM-ISMR- and MMM-NINO3.4-indices seems not very meaningful. As another example, I am not sure, the premise of the paragraph on Page 6 Line 8 is valid.

Response:

Thank you for the comment. Accordingly, we have removed the relevant MMM correlations.

Concern:

You often refer to proxies but don't show any comparisons at all. For example on page 12 line 3, it would help to be able to make the comparison directly or at least in an appendix.

Response:

We had discussed related proxy records for comparisons in citations (e.g. Figure 8 of Ramesh et al., 2011; Box TS.5, Figure-1 of Technical Summary in the IPCC 2013; Stocker 2013), few more we will add in appendix part of revision.

Concern:

P7L14 and L16: I do not see the decreasing precipitation trend. If it is there, please quantify it more clearly.

Response:

Thank you. We have removed the statement.

Concern:

P12L7: I'd like to repeat, that I don't think these correlations mean much for the multi-model-ensemble-mean, as you don't identify the relation between ENSO and Indian climate by this, but just the common long term relation between the global mean and both the ENSO-region and India.

Additionally, the correlation coefficients are generally rather small, which to me also indicates that, at least for temperature, you just capture the concurrent relation between TG and TI and TG and NINO3.4.

Response:

As we had mentioned earlier, we have purged the discussion of the ISMR-NINO index analysis relevant to the MMM.

We have carried out the partial correlation between TG, TI and NINO3.4 to remove the concurrent TG signal on TI (not shown). This analysis demonstrates the significant impact of the ENSO on TI temperature at a 0.05 confidence level.

Concern:

Furthermore, you make some additional inferences which I would call wrong or at least your analyses do not warrant them, e.g.:

P11L25: "All this suggests a better agreement among the models in simulating the long term trend in rainfall over India relative to the variability during the MWP and that during the LIA."

I don't think your analyses allow to infer this.

Response:

Thank you for the comment. We removed the statement.

Concern:

P13L10: "we surmise that Indian Sub-continent was warmer (relatively cooler) during the CE 1000-1199 (CE 1550-1749) than the many other regions of the globe".

I don't understand how you come to this assumption, but I don't think you can infer this from comparing one regional series with the global mean.

Response:

We modify this sentence to "We surmise that Indian Sub-continent was warmer (relatively cooler) during the CE 1000-1199 (CE 1550-1749) than the concurrent global mean temperature"

Concern:

P14L3: "More importantly, these results suggest that any long term weakening between the ENSO-Indian summer monsoon (e.g. Kumar et al., 1999; Ashok et al., 2001) is not necessarily due to anthropogenic climate change." It's not that this isn't a possibility but from my point of view your data and your analyses do not give any indication of this, they in my opinion don't tell you anything at all about this.

Response:

We have re written the text for better clarity. Now it reads as "we can make a conjecture that any weakening between the ENSO-Indian summer monsoon (e.g. Kumar et al., 1999), associated with changed variability of either of them (Kriplani et al., 1999; Ashok et al., 2007), or for that matter associated with changed strength or frequency of another monsoon-driver such as the IOD (e.g. Ashok et al., 2001), may not be necessarily due to anthropogenic climate change."

Concern:

[Figure]

P14L21: "It is known that the El Niños (La Niñas) cause anomalous increase (decrease) in global temperature. Therefore, a predominant presence of higher number of simulated El Niños as compared to La Niñas in almost all the models is the reason why the simulated MWP is warmer as compared to the LIA. Given this agreement across the models, we can surmise that, in real world too, the MWP is likely due to the occurrence of a relatively higher frequency of El Niños as compared to the La Niñas"

a) Can you give references for "It is known . . ."

Response:

Trenberth et al., (2002)

Concern:

b) You do not identify a change in frequency of El Ninos but just describe the temperature change in the background state.

Response:

We have refined the frequency distinction analysis for the sake of a better objectivity, notwithstanding that the result is qualitatively the same as compared to that from the earlier version of the manuscript. Specifically, the earlier version, we had identified a simulated ENSO event by fixing an amplitude-threshold of $0.5°C$ for the Nino3.4 index. Now, we identify a simulated ENSO event when the amplitude of the NINO3.4 index exceeds its standard deviation (If temperature is below one standard deviation considered as El Niño (La Nina) and if it is above one standard deviation, it is considered as strong El Niño or strong La Niña events.

We still find that there are relatively more El Ninos (La Ninas) during simulated MWP (LIA) as compared to LIA (MWP), and conjecture that such a frequency distinction may have a role in with the relatively warmer (cooler) MWP (LIA). Having said that, this hypothesis needs some specialised sensitivity experiments with AGCMs, which we plan to, do in near future. Based on the statistically significant NINO3.4-ISMR correlations,

which are similar to those from current day observations, during both MWP & LIA, we believe that the interannual teleconnection impacts have been somewhat reduced by the long term changes. All these details also go into the revision.

Concern:

c) That is, the MWP in the models isn't warmer and LIA isn't colder because of ENSO. Rather your analysis just describes the warmer MWP and the colder LIA. d) Thus your inference is circular and in this context very likely wrong. Response:

For better clarity, we explain the limitations. Please see the above response.

Concern:

P19L2: "Our analysis of the PMIP3 datasets suggests that the Indian region was likely warmer than the global temperature during the MWP."

I don't think your analysis suggests this.

Response:

We have compared the global spatial anomaly of surface temperatures with Indian surface temperatures anomaly which clarifies this aspect. Spatial plots of global and Indian region anomalous temperatures have been attached. (Figures R3)

Minor points:

Concern:

Is the title appropriate? I would say, you discuss more the ISM-ENSO relation than the general state of the ISM.

Response:

Thanks for suggestion. We now modify the title slightly to "The Indian summer monsoon climate and its connection to ENSO through the Last Millennium, as simulated by the PMIP3"

Concern:

I would suggest to drop the MIROC-simulation completely from the paper. It is enough to mention from the start that you don't use it, because of the known problems with the simulation. There are a number of references available, I think.

Response:

We have accordingly dropped the MIROC from our analysis from both Historical and LM.

Concern:

I found the manuscript in parts hard to read, thus I would suggest to check where things could be rewritten to make the manuscript more clearly readable.

Response:

Thank you. We have revised the manuscript carefully to make it more clearly & better readable.

Concern:

Page 1 Line 24: The models can't confirm the proxy-data, they can only be consistent with them or, if you insist, can agree with the proxies.

Response:

We have modified the sentence accordingly, by replacing "in confirmation" to "consistent with"

Concern:

P2L11: What do you mean by "there is no apparently significant change in the external climate forcing from the first half of 20th century"?

Response:

[Figure]

It means, there is no apparent change in external forcing (solar forcing).

Concern:

P3L5: Is this full paragraph relevant for this paper?

Response:

We believe so, as this paragraph essentially, but just briefly, sum up the few proxy-based papers that discuss the Indian summer monsoon rainfall changes between the MWP & LIA.

Concern:

P3L33: Please give a reference for PMIP3.

Response:

Thank you. We cited it properly now with Schmidt et al. (2012).

Concern:

P4L19: I don't think you validate the models

Response:

We validate simulated features such as the temperature trends, ISMR-ENSO links, etc., in the historical simulations by the models by comparing these features with those from the available observed and reanalysis data sets. The actual paragraph reads as "It is indeed a challenging prospect to validate the simulated Indian summer monsoon features from the PMIP3 simulations for the LM period given the sparse and scanty observations. Fortunately, model simulations of the CMIP5 for the historical period (CE 1850-2005) can be validated using various observed/reanalysed gridded datasets, keeping in mind the uncertainties associated with such datasets during the pre-satellite period"

Concern:

P4L33: The correct references for the PMIP3-past1000 simulation-setup are Schmidt et al. (2011, doi:10.5194/gmd-4-33-2011) and Schmidt et al. (2012, doi:10.5194/gmd-5-185-2012)

Response:

Thank you for pointing out. We have cited these in the revised manuscript.

Concern:

P5L6: I don't think ERA-Interim goes from 1900 to 2010?

Response:

Sorry for the mistake. It should read as ERA-20CM (Monthly means of daily means). We have now revised it.

Concern:

P5L8: Can you please provide a reference for the IMD-data? And is this data publically available? If yes, please provide a URL, if not please provide contact details where the data can be obtained. [Dear editor, please crosscheck this with the data availability requirements.]

Response:

We did mention the reference citation. It is Rajeevan et al., (2006).

Concern:

P6L33: Is this relevant? What does this imply?

Response:

In the manuscript, Figures 1c and 1d show anomalous global and Indian surface temperatures with 11-year running mean. We had shown this to show that all the models are able to capture the current warming trend.

Concern:

P7L11: You do not plot events but just the 11-running means which possibly masks the higher frequent ENSO-variability.

Response:

We have corrected it (removed the relevant smoothed time series of the Nino3.4 index with the unsmoothed time series). The Nino3.4 frequency tables also clearly show the higher (lower) number of simulated warm (cold) ENSO events, represented by the positive Nino3.4 index, during the MWP (LIA)"

Concern:

P8L18: You shortly write about standard deviations here and elsewhere. From my point of view, it does not become clear, what's the point of these discussions.

Response:

This analysis has been carried out to show the spread of any parameter mentioned across the models.

Concern:

P10L16: Which models are these outliers, what is the bias because of which you call them outliers, where do you show this?

Response:

The models whose anomalous TG is above or below $1\sigma$ are outliers. In general, that the spread in various statistics across the models is within the limits defined by $1\sigma$ except the model S2.

Concern:

P11L2: Much of what you describe here for the global temperature is in the IPCC and other publications.

Response:

We agree. We had actually mentioned so in the manuscript.

Concern:

P11L18: I do not see these decreasing ISMR-trends, please clarify.

Response:

For better clarity we have mentioned a trend line diagram of simulated ISMR during LM (Figure R4) below with the trend line equations. We have modified the sentence for clarity.

Concern:

P11L20: I cannot really follow the premise of this sentence.

Response:

We have modified the sentence for better clarity.

Concern:

P11L22: Why would we "expect" this?

Response:

In general, slowly increasing temperatures in the tropical regions are associated with a decreasing rainfall. This is based on the Tetens formula which suggests that every 1C rise in temperature leads to the moisture holding capacity of the atmosphere by 7%.

Concern:

P12L11: I think you mean "1000-year".

Response:

Corrected the typo.

Concern:

P13L1: Maybe I miss something why are these statistics of interest/relevant?

Response:

Standard deviation analysis has been carried out to show that the spread of any parameter mentioned across the models.

Concern:

P13L17: What do you mean by "realistic"?

Response:

Thank you. We revise the sentence to "All these correlations are comparable to the corresponding correlations from observations during the historical period, as well as ....."

Concern:

P13L19: GOALS should read FGOALS.

Response:

Corrected it.

Concern:

Your Figure 7 does not show the ensemble but only one model, as far as I can see. Response:

Thank you. Corrected the typo.

Concern:

P15L4: What do you mean by this "discrepancy".

Response:

We revised this sentence for better clarity as "Further, there is relatively more discrepancy in the simulated El Niño & La Niña frequencies, i.e. the skewness of ENSO, across the models in the LIA simulations as compared to those for the MWP.

Concern:

P15L5: What kind of "factor" is this "discrepancy" meant to be?

Response:

We realise that this sentence is ambiguous. We remove it.

Concern:

P17L7: Maybe you should discuss the different models first before writing about the composite.

Response:

In the revision, we have briefly listed out the details of the models in a table.

Concern:

P19L29: Can you provide a reference for this cautionary note.

Response:

We have already provided some references. The sentence ("A plausible. . .") is basically speculative one.

Concern:

P20L8: Maybe I missed it in your results-section but I think you should discuss these contrasts between the LM-relations and the modern relations in more detail and possibly show them in Figures or at least supplementary materials.

Response:

We will discuss more about individual in revised version.

Concern:

The Figures produced with GrADS are sometimes of suboptimal quality. Furthermore I recommend to change the color scale which is rather easy in GrADS if I recall correctly. The reason for this is, e.g., https://www.climate-lab-book.ac.uk/2014/end-of-the-rainbow/.

Response:

Our sincere apologies. We have changed the colour scale.

Concern:

It is often unclear whether MMM refers to the multi-model-mean or to the multi-model-ensemble, its members, or its spread. One example is on page 8 in line 11.

Response:

MMM refers to multi-model mean not multi model ensemble.

Concern:

Please do not insert tables as pictures into the manuscript.

Response:

Thank you, we corrected it.

Please also note the supplement to this comment:
https://www.clim-past-discuss.net/cp-2017-24/cp-2017-24-AC1-supplement.pdf

[Figure]

[Figure]

Figure R1: Correlation between simulated JJAS NINO3.4 and simulated JJAS surface temperatures
zoomed over Indian region during MWP and LIA.

**Fig. 1.**

[Figure]

Figure R2: Correlation between simulated JJAS NINO3.4 and simulated ISMR during MWP and LIA.

**Fig. 2.**

[Figure]

*Figure R3: Spatial distribution of anomalous surface temperatures during MWP and LIA.*

**Fig. 3.**

[Figure]

Figure R4: Linear trend plot of ISMR during LM.

**Fig. 4.**

**Supplement:**

**Responses to Reviewer 1**

*At the outset, we would like to thank the reviewer for the useful and encouraging comments, which have improved the standard of the manuscript.*

**Main concerns:**

*Question 1:* **First, you diagnose differences in correlation coefficients between two periods. However, Gershunov et al. (2001, http://dx.doi.org/10.1175/1520-442(2001)014<2486:LFMOTE>2.0.CO;2) and others discuss how correlation coefficients can change over time without a change in the underlying relation. My impression is, that maybe one if any correlation-difference may be significant in your analyses.**

*Response:*

Thank you for the comment. Gershunov et al., 2001 paper cautions about the sampling issues associated with a short (21-31year) running correlation within a 130-150 yrs period, which does not vary in terms of external forcings (i.e. change in solar signal, or significant volcanic eruptions etc.). In such a case (specifically, weakening of ENSO-Indian summer monsoon rainfall in the last 15-20years of the $20^{th}$ century), it is very important to check if any such change in decadal variability may be subject to stochasticity.

However, in our case, we evaluate changes in a statistic (correlation) between two separate climatic (200 year) regimes, with different external forcings such as different frequencies of volcanic eruption, etc. Therefore, our results are not subject to the Gershunov et al., 2001 concern.

As we have shown in the manuscript, the difference between the ENSO-ISMR correlation from the MWP simulations and the LIA simulations are statistically significant at 0.05 level from a Student's two tailed t-test.

Having said this, we have anyway carried out a bootstrapping significance test for the above mentioned difference in correlation (1000 simulations). We find that the results from 4 out of 7 models are significant at 95% confidence level.

*Question* **2: Secondly, if I understand your analyses of the ENSO-frequencies correctly, you do not identify differences in La Nina and El Nino frequencies but just diagnose differences in the mean background sta**te.

*Response:*

Thank you.  Actually, we had shown the difference in the frequencies of El Ninos and La Ninas to explain the relatively warm MWP and relatively cold LIA temperatures. Specifically, the MWP simulations exhibit more number of El Ninos relative to the La Ninas. It is well known that the El Ninos (La Ninas) result in warmer (cooler) than normal global temperatures (e.g. Trenberth et al., 2002). Therefore, the simulated above normal global temperatures during the MWP are in conformation with the relatively more number of simulated El Ninos. The simulated relatively cooler LIA is also easily explained by the simulated ENSO frequency skewed negatively, that is more simulated La Ninas as compared to El Ninos during this period. Our results also clearly indicate (Figure R1) that the simulated global mean temperature is positively and significantly

correlated with the Nino3.4 index during the LM and its sub-periods.

The differences in the simulated mean background state shown, on the other hand, explain why the simulated ISMR during the MWP is more than that during the LIA. For example, the 'anomalies' in large scale convergence and divergence patterns during the MWP relative to the LM averages explain that there is a relatively eastward shift in Walker circulation, with associated changes over India, which have resulted in more rainfall during this period as compared to the LM. To be clear, the correlations of the simulated Nino3.4 index with local summer monsoon temperatures over various places in India (Figure-A4 of the revised manuscript, attached below as Figure R1), and that with the ISMR (Figure-A5, attached below as Figure R2) are all negative, as is observed in general during the last 150years or so.

We must mention that, in the revision, we have refined the frequency distinction analysis for the sake of a better objectivity, notwithstanding that the result is qualitatively the same as compared to that from the earlier version of the manuscript. Specifically, the earlier version, we had identified a simulated ENSO event by fixing an amplitude-threshold of 0.5℃ for the Nino3.4 index. Now, we identify a simulated ENSO event when the amplitude of the NINO3.4 index exceeds its standard deviation (If temperature is below one standard deviation considered as La Nina and if it is above one standard deviation, it is considered as strong El Nino or La Nina events). We still find that there are relatively more El Ninos (La Ninas) during simulated MWP (LIA) as compared to LIA (MWP).

*Trenberth, K. E., J. M. Caron, D. P. Stepaniak, and S. Worley, Evolution of El Niño–Southern Oscillation and global atmospheric surface temperatures, J. Geophys. Res., 107(D8), doi:10.1029/2000JD000298, 2002.*

**Further concerns:**

I am not sure your use of the Multi-Model-mean is appropriate. I think for most of your analyses the spread and the differences between the models are of interest and the multi-model-mean analysis is not necessary. For example correlating the MMM-ISMR-and MMM-NINO3.4-indices seems not very meaningful. As another example, I am not sure, the premise of the paragraph on Page 6 Line 8 is valid.

Thank you for the comment. Accordingly, we have removed the relevant MMM correlations.

You often refer to proxies but don't show any comparisons at all. For example on page 12 line 3, it would help to be able to make the comparison directly or at least in an appendix.

We had discussed related proxy records for comparisons in citations (e.g. Figure 8 of Ramesh et al., 2011; Box TS.5, Figure-1 of Technical Summary in the IPCC 2013; Stocker 2013), few more we will add in appendix part of revision.

P7L14 and L16: I do not see the decreasing precipitation trend. If it is there, please quantify it more clearly.

Thank you. We have removed the statement.

P12L7: I'd like to repeat, that I don't think these correlations mean much for the multi-model-ensemble-mean, as you don't identify the relation between ENSO and Indian climate by this, but just the common long term relation between the global mean and both the ENSO-region and India.

**Additionally, the correlation coefficients are generally rather small, which to me also indicates that, at least for temperature, you just capture the concurrent relation between TG and TI and TG and NINO3.4.**

As we had mentioned earlier, we have purged the discussion of the ISMR-NINO index analysis relevant to the MMM.

We have carried out the partial correlation between TG, TI and NINO3.4 to remove the concurrent TG signal on TI (not shown). This analysis demonstrates the significant impact of the ENSO on TI temperature at a 0.05 confidence level.

**Furthermore, you make some additional inferences which I would call wrong or at least your analyses do not warrant them, e.g.:**

**P11L25: "All this suggests a better agreement among the models in simulating the long term trend in rainfall over India relative to the variability during the MWP and that during the LIA."**

**I don't think your analyses allow to infer this.**

Thank you for the comment. We removed the statement.

**P13L10: "we surmise that Indian Sub-continent was warmer (relatively cooler) during the CE 1000-1199 (CE 1550-1749) than the many other regions of the globe".**

**I don't understand how you come to this assumption, but I don't think you can infer this from comparing one regional series with the global mean.**

We modify this sentence to "We surmise that Indian Sub-continent was warmer (relatively cooler) during the CE 1000-1199 (CE 1550-1749) than the concurrent global mean temperature"

**P14L3: "More importantly, these results suggest that any long term weakening between the ENSO-Indian summer monsoon (e.g. Kumar et al., 1999; Ashok et al., 2001) is not necessarily due to anthropogenic climate change."**
**It's not that this isn't a possibility but from my point of view your data and your analyses do not give any indication of this, they in my opinion don't tell you anything at all about this.**

We have re written the text for better clarity. Now it reads as "*we can make a conjecture that any weakening between the ENSO-Indian summer monsoon (e.g. Kumar et al., 1999), associated with changed variability of either of them (Kriplani et al., 1999; Ashok et al., 2007), or for that matter associated with changed strength or frequency of another monsoon-driver such as the IOD (e.g. Ashok et al., 2001), may not be necessarily due to anthropogenic climate change.*"

**P14L21: "It is known that the El Niños (La Niñas) cause anomalous increase (decrease) in global temperature. Therefore, a predominant presence of higher number of simulated El Niños as compared to La Niñas in almost all the models is the reason why the simulated MWP is warmer as compared to the LIA. Given this agreement across the models, we can surmise that, in real world too, the MWP is likely due to the occurrence of a relatively higher frequency of El Niños as compared to the La Niñas"**

**a) Can you give references for "It is known . . ."**

Trenberth et al., (2002)

**b) You do not identify a change in frequency of El Ninos but just describe the temperature change in the background state.**

We have refined the frequency distinction analysis for the sake of a better objectivity, notwithstanding that the result is qualitatively the same as compared to that from the earlier version of the manuscript. Specifically, the earlier version, we had identified a simulated ENSO event by fixing an amplitude-threshold of 0.5℃ for the Nino3.4 index. Now, we identify a simulated ENSO event when the amplitude of the NINO3.4 index exceeds its standard deviation (If temperature is below one standard deviation considered as El Nino (La Nina) and if it is above one standard deviation, it is considered as strong El Nino or strong La Nina events.

We still find that there are relatively more El Ninos (La Ninas) during simulated MWP (LIA) as compared to LIA (MWP), and conjecture that such a frequency distinction may have a role in with the relatively warmer (cooler) MWP (LIA). Having said that, this hypothesis needs some specialised sensitivity experiments with AGCMs, which we plan to, do in near future. Based on the statistically significant NINO3.4-ISMR correlations, which are similar to those from current day observations, during both MWP & LIA, we believe that the interannual teleconnection impacts have been somewhat reduced by the long term changes. All these details also go into the revision.

**c) That is, the MWP in the models isn't warmer and LIA isn't colder because of ENSO. Rather your analysis just describes the warmer MWP and the colder LIA. d) Thus your inference is circular and in this context very likely wrong.**

For better clarity, we explain the limitations. Please see the above response.

**P19L2: "Our analysis of the PMIP3 datasets suggests that the Indian region was likely warmer than the global temperature during the MWP."**

**I don't think your analysis suggests this.**

We have compared the global spatial anomaly of surface temperatures with Indian surface temperatures anomaly which clarifies this aspect. Spatial plots of global and Indian region anomalous temperatures have been attached. (Figures R3)

**Minor points:**

**Is the title appropriate? I would say, you discuss more the ISM-ENSO relation than the general state of the ISM.**

Thanks for suggestion. We now modify the title slightly to "The Indian summer monsoon climate and its connection to ENSO through the Last Millennium, as simulated by the PMIP3"

**I would suggest to drop the MIROC-simulation completely from the paper. It is enough to mention from the start that you don't use it, because of the known problems with the simulation. There are a number of references available, I think.**

We have accordingly dropped the MIROC from our analysis from both Historical and LM.

**I found the manuscript in parts hard to read, thus I would suggest to check where things could be rewritten to make the manuscript more clearly readable.**

Thank you. We have revised the manuscript carefully to make it more clearly & better readable.

**Page 1 Line 24: The models can't confirm the proxy-data, they can only be consistent with them or, if you insist, can agree with the proxies.**

We have modified the sentence accordingly, by replacing "in confirmation" to "consistent with"

**P2L11: What do you mean by "there is no apparently significant change in the external climate forcing from the first half of 20th century"?**

It means, there is no apparent change in external forcing (solar forcing).

**P3L5: Is this full paragraph relevant for this pap**er?

We believe so, as this paragraph essentially, but just briefly, sum up the few proxy-based papers that discuss the Indian summer monsoon rainfall changes between the MWP & LIA.

**P3L33: Please give a reference for PMIP3.**

Thank you. We cited it properly now with Schmidt et al. (2012).

**P4L19: I don't think you validate the models**

We validate simulated features such as the temperature trends, ISMR-ENSO links, etc., in the historical simulations by the models by comparing these features with those from the available observed and reanalysis data sets. The actual paragraph reads as "It is indeed a challenging prospect to validate the simulated Indian summer monsoon features from the PMIP3 simulations for the LM period given the sparse and scanty observations. Fortunately, model simulations of the CMIP5 for the historical period (CE 1850-2005) can be validated using various observed/reanalysed gridded datasets, keeping in mind the uncertainties associated with such datasets during the pre-satellite period"

**P4L33: The correct references for the PMIP3-past1000 simulation-setup are Schmidt et al. (2011, doi:10.5194/gmd-4-33-2011) and Schmidt et al. (2012, doi:10.5194/gmd-5-185-2012)**

Thank you for pointing out. We have cited these in the revised manuscript.

**P5L6: I don't think ERA-Interim goes from 1900 to 2010?**

Sorry for the mistake. It should read as ERA-20CM (Monthly means of daily means). We have now revised it.

**P5L8: Can you please provide a reference for the IMD-data? And is this data publically available? If yes, please provide a URL, if not please provide contact details where the data can be obtained. [Dear editor, please crosscheck this with the data availability requirements.]**

We did mention the reference citation. It is Rajeevan et al., (2006).

**P6L33: Is this relevant? What does this imply?**

In the manuscript, Figures 1c and 1d show anomalous global and Indian surface temperatures with 11-year running mean. We had shown this to show that all the models are able to capture the current warming trend.

**P7L11: You do not plot events but just the 11-running means which possibly masks the higher frequent ENSO-variability.**

We have corrected it (removed the relevant smoothed time series of the Nino3.4 index with the unsmoothed time series). The Nino3.4 frequency tables also clearly show the higher (lower) number of simulated warm (cold) ENSO events, represented by the positive Nino3.4 index, during the MWP (LIA)"

**P8L18: You shortly write about standard deviations here and elsewhere. From my point of view, it does not become clear, what's the point of these discussions.**

This analysis has been carried out to show the spread of any parameter mentioned across the models.

**P10L16: Which models are these outliers, what is the bias because of which you call them outliers, where do you show this?**

The models whose anomalous TG is above or below $1\sigma$ are outliers. In general, that the spread in various statistics across the models is within the limits defined by $1\sigma$ except the model S2.

**P11L2: Much of what you describe here for the global temperature is in the IPCC and other publications.**

We agree. We had actually mentioned so in the manuscript.

**P11L18: I do not see these decreasing ISMR-trends, please clarify.**

For better clarity we have mentioned a trend line diagram of simulated ISMR during LM (Figure R4) below with the trend line equations. We have modified the sentence for clarity.

**P11L20: I cannot really follow the premise of this sentence.**

We have modified the sentence for better clarity.

**P11L22: Why would we "expect" this?**

In general, slowly increasing temperatures in the tropical regions are associated with a decreasing rainfall. This is based on the Tetens formula which suggests that every 1C rise in temperature leads to the moisture holding capacity of the atmosphere by 7%.

**P12L11: I think you mean "1000-year".**
Corrected the typo.
**P13L1: Maybe I miss something why are these statistics of interest/relevant?**
Standard deviation analysis has been carried out to show that the spread of any parameter mentioned across the models.

**P13L17: What do you mean by "realistic"?**

Thank you. We revise the sentence to "All these correlations are comparable to the corresponding correlations from observations during the historical period, as well as ....."

**P13L19: GOALS should read FGOALS.**

Corrected it.

**Your Figure 7 does not show the ensemble but only one model, as far as I can see.**

Thank you. Corrected the typo.

**P15L4: What do you mean by this "discrepancy".**

We revised this sentence for better clarity as "Further, there is relatively more discrepancy in the simulated El Niño & La Niña frequencies, i.e. the skewness of ENSO, across the models in the LIA simulations as compared to those for the MWP.

**P15L5: What kind of "factor" is this "discrepancy" meant to be?**
We realise that this sentence is ambiguous. We remove it.

**P17L7: Maybe you should discuss the different models first before writing about the composite.**

In the revision, we have briefly listed out the details of the models in a table.

**P19L29: Can you provide a reference for this cautionary note.**

We have already provided some references. The sentence ("A plausible…") is basically speculative one.

**P20L8: Maybe I missed it in your results-section but I think you should discuss these contrasts between the LM-relations and the modern relations in more detail and possibly show them in Figures or at least supplementary materials.**

We will discuss more about individual in revised version.

**The Figures produced with GrADS are sometimes of suboptimal quality. Furthermore I recommend to change the color scale which is rather easy in GrADS if I recall correctly. The reason for this is, e.g., https://www.climate-lab-book.ac.uk/2014/end-of-the-rainbow/.**

Our sincere apologies. We have changed the colour scale.

**It is often unclear whether MMM refers to the multi-model-mean or to the multi-model-ensemble, its members, or its spread. One example is on page 8 in line 11.**

MMM refers to multi-model mean not multi model ensemble.

**Please do not insert tables as pictures into the manuscript.**

Thank you, we corrected it.

[Figure]

*Figure R1: Correlation between simulated JJAS NINO3.4 and simulated JJAS surface temperatures*

[Figure]

*zoomed over Indian region during MWP and LIA.*

*Figure R2: Correlation between simulated JJAS NINO3.4 and simulated ISMR during MWP and LIA.*

[Figure]

*Figure R3: Spatial distribution of anomalous surface temperatures during MWP and LIA.*

[Figure]

*Figure R4: Linear trend plot of ISMR during LM.*

---

## Author Comment (AC2) · 18 Aug 2017

Thank you for the comment. CP is doing a great job by including the Data Citations; this would help expanding the data availability to the scientific community.

Data:

In our study, we used CMIP5/PMIP3 Last Millennium (LM) simulations and corresponding Historical simulations (HS) data with few observational/reanalysis data.

Simulations of CMIP5/PMIP3 LM and HS can be found at public repository and available to the scientific community. <https://cera-www.dkrz.de/WDCC/ui/cerasearch/>.

Reanalysis/observational datasets been used in our study are collected from NCEP <https://www.esrl.noaa.gov/psd/data/gridded/data.ncep.reanalysis.html>, ECMWF <https://apps.ecmwf.int/datasets/data/era20cm-edmm/levtype=sfc/> and Indian meteorological centre <https://www.tropmet.res.in/static_page.php?page_id=52>, we have mentioned proper citations for each of them and they are available for scientific community from respective public archives.

Our analysis data:

We would be happy to provide our analysis data, figures, excel sheets and tables through a public repository or make them available through Research gate or Google drive to scientific community. We will provide detailed information of our analysis with a link to the archive in revision of our current manuscript in Data Availability section.

---

## Author Comment (AC3) · 18 Aug 2017

Thank you for identifying the typing mistake. The NINO3.4 region we used in our study, indeed, is 170°W-120°W; 5°S-5°N. We corrected both the tying mistake.

---

## Author Comment (AC4) · 18 Aug 2017

At the outset, we would like to thank the reviewer for her useful and encouraging comments, which have improved the standard of the manuscript.

Specific Comments:

(1) Paleoclimate reconstructions from proxy data suggest that during the MWP, a cooler tropical eastern Pacific, referred to as a La Niña-like background state, is reconstructed. However, this is not evident in the PMIP3 model simulations. Almost all the models except one consistently simulate more El Niños as compared to La Niñas during the

MWP compared to the LIA.

In P.14, the authors state that "It is known that the El Niños (La Niñas) cause anomalous increase (decrease) in global temperature. Therefore, a predominant presence of higher number of simulated El Niños as compared to La Niñas in almost all the models is the reason why the simulated MWP is warmer as compared to the LIA. Given this agreement across the models, we can surmise that, in real world too, the MWP is likely due to the occurrence of a relatively higher frequency of El Niños as compared to the La Niñas".

I think this statement is not reasonable. The reconstruction exhibits a La Niña-like pattern in the tropical Pacific during the MCA (Cobb et al. 2003; Graham et al. 2007; Mann et al. 2009). Besides, the La Niña-like condition is reproduced in simulations employing the simplified Zebiak-Cane model of the tropical Pacific coupled oceanatmosphere system (Mann et al. 2005), which exhibits a stronger dynamical feedback than most global models. Thus, it is not correct to say that in real world, the MWP is likely due to the occurrence of a relatively higher frequency of El Niños as compared to the La Niñas" just from the perspective of model results. The global temperature changes may have been driven by the effective radiative forcing during the past millennium. However, there is little evidence for globally synchronized MCA and LIA intervals, with the specific timing of these intervals varying regionally, which may have been dominated by the internal variability.

Cobb, K., C. Charles, H. Cheng, and R. Edwards, 2003: El Nino/Southern Oscillation and tropical Pacific climate during the last millennium. Nature, 424, 271- 276. Graham, N. E., and Coauthors, 2007: Tropical Pacific-mid-latitude teleconnections in medieval times, Climatic Change, 83, 241-285. Mann, M. E., M. A. Cane, S. E. Zebiak, and A. Clement, 2005: Volcanic and solar forcing of the tropical Pacific over the past 1000 years. J. Climate, 18, 447-456. Mann, M. E., and Coauthors, 2009: Global signatures and dynamical origins of the Little Ice Age and Medieval Climate Anomaly. Science, 326, 1256-1260.

Response:

Thank you for thought-provoking comment. Motivated by the suggestion, we checked the suggested papers out, and in the process, found another recent paper. While the papers suggested by the reviewer suggest that MCA has been a host to more La Niñas, there is at least one proxy-based paper, Conroy et al., (2008), which finds that their diatom record are not consistent on SST interpretation with that of a coral record (Cobb et al., 2003). Specifically, while the diatom record suggests warmer SST in the eastern equatorial pacific during a portion of the medieval period, the coral derived SST indicates a cooling trend. Conroy et al. (2008) suggest a more heterogeneous SST in the region.

Therefore, we temper our discussion as follows.

"Interestingly, a majority of the PMIP3 models in this study indicate more El Niños as compared to the La Niñas during the MWP. Tellingly, in the recent period, El Niños (La Niñas) have been suggested to cause anomalous increase (decrease) in global temperature (e.g. Trenberth and Stepaniak, 2001). Therefore, a predominant presence of higher number of simulated El Niños as compared to La Niñas in almost all the models is a possible reason why the simulated MWP, at least in some tropical regions, is warmer as compared to the LIA. Having said this, this needs to be verified by making some AGCM sensitivity experiments (such as forcing them with the MWP & LIA SSTs, and later repeat them by removing a few El Niño/La Niñas), which we plan to do I near future.

Importantly, a study using a Cane-Zebiak type of coupled model (Mann et al., 2005) suggests more La Niña-like conditions during the MWP. Several proxy-data studies (Cobb et al. 2003; Graham et al. 2007; Mann et al. 2009) suggest either a weak ENSO variance or more La Niñas during the MWP. Then again, a study by Conroy et al., (2008), which finds that their diatom record is not consistent on SST interpretation with that of a coral record (Cobb et al., 2003). Specifically, while the diatom record

suggests warmer SST in the eastern equatorial pacific during a portion of the medieval period, the coral derived SST indicates a cooling trend. Conroy et al. (2008) suggest a more heterogeneous SST in the region. Therefore, a predominant presence of higher number of simulated El Niños as compared to La Niñas in almost all the models is the reason why the simulated MWP is warmer as compared to the LIA. Given this agreement across the models, which have a more detailed oceanic component as compared to that used in Mann et al. (2005), we can surmise that, in real world too, the possibility of occurrence of MWP due to the occurrence of a relatively higher frequency of El Niños as compared to the La Niñas cannot be ruled out completely".

Conroy et al. (2008). Unprecedented recent warming of surface temperatures in the eastern tropical Pacific Ocean. Nature Geoscience 2, 46 - 50 (2009) doi:10.1038/ngeo390

(2) The simulated ISMR anomaly shows a weak decreasing trend throughout the LM. The authors also attributed the possible dynamics to the more number of El Niños during the MWP as compared to the LM. The distribution of summer velocity potential at 850 hPa suggests a westward shift in Walker circulation, and the anomalous divergence center in the west also extends into the equatorial eastern Indian Ocean, which results in an anomalous convergence zone over India and therefore excess rainfall during the MWP. It is good that the model results are inter-consistent by themselves. Proxy records also suggest that the ISMR was higher during the MWP and relatively weaker during the LIA (Yadava et al. 2005). A speleothem-based reconstruction of ISMR variability exhibits an increased summer monsoon precipitation during the MWP and a severe weakening of monsoon rainfall during the LIA, apparently associated with droughts particularly between 13th and 17th centuries. However, proxy reconstructions show opposite ENSO conditions as compared with the simulations during the MWP and LIA periods, how can we explain the ENSO-monsoon relationship and the possible dynamics from the reconstruction perspective?

Response:

Thank you for the comment. The response to the above comment provides a partial answer in relation to the uncertainties associated with the ENSO conditions. In addition, models are in confirmation with proxy studies such as Yadava et al., Ramesh et al., Thamban et al. in the sense that they simulate above normal (below) rainfall during MWP (LIA).

We have ascertained through a correlation analysis between Nino3.4 index and 850 hPa Velocity Potential that many models reproduce the typical anomalous divergence over the western pacific-through-Indian region; associated with El Niños (the figure below shows some samples). From this, it is apparent that while the interannual dynamics behind ENSO-ISMR connections are likely similar to that during the current day, the long term changes in the Walker circulation between the MWP and LIA modulate the El Niñoo impacts.

(3) Apart from the Walker circulation changes, does the land-sea thermal contrast change in the upper-troposphere also play an important role for the ISMR variability during the LM? If yes, can we further attributed to the external forcing drivers? Since the correlations between ENSO and the ISMR may differ on the multi-decadal-to-centennial scales from that on the inter-annual timescales.

Response:

Thank you for the very important comment. We have checked the simulated land sea thermal gradient. Our new analysis shows a weakening land sea gradient at the 850 hPa (e.g. Sinha et al., 2015; Roxy et al., 2015) during LIA compared to MWP in five out of eight models, and also in the upper troposphere (e.g. Goswami et al., 2006; Wang et al., 2013). . Having said that, it is difficult to say whether this is related to the decadal circulation changes associated with ENSO, or independent of them. We cannot also comment whether such changes are associated with external forcings such as volcanoes, unless we conduct sensitivity experiments with AGCMs. Unfortunately, carrying out such experiments is beyond the scope of the current study. These aspects

are also reflected in the revised text.

Just for information, Eurasian snow cover, land sea contrast, Atlantic variability, etc. are some of the other forcings suggested in addition to the tropical Indo-pacific drivers.

Sinha, A. et al. Trends and oscillations in the Indian summer monsoon rainfall over the last two millennia. Nat. Commun. 6:6309 doi: 10.1038/ncomms7309 (2015).

Goswami, B. N., Madhusoodanan, M. S., Neema, C. P. & Sengupta, D. A physical mechanism for North Atlantic SST influence on the Indian summer monsoon. Geophys. Res. Lett. 33, L02706 (2006).

Wang et al., 2013. Northern Hemisphere summer monsoon intensified by mega-El Nino/southern oscillation and Atlantic Multidecadal oscillation. Proc. Natl. Acad. Sci. USA 110, 5347–5352.

Typing Errors:

(1) P.16, Line 16, there are two "due to", delete one Response: Thank you. Corrected it.

Please also note the supplement to this comment:
https://www.clim-past-discuss.net/cp-2017-24/cp-2017-24-AC4-supplement.pdf

---

## Author Comment (AC5) · 19 Aug 2017

Modified response to question 3 of Reviewer 2. Consider this new response of question 3 and previous response may be discarded. Sorry for the inconvenience happened.

(3) Apart from the Walker circulation changes, does the land-sea thermal contrast change in the upper-troposphere also play an important role for the ISMR variability during the LM? If yes, can we further attributed to the external forcing drivers? Since the correlations between ENSO and the ISMR may differ on the multi-decadal-to-centennial scales from that on the inter-annual timescales.

[Figure]

Response:

Thank you for the very important comment. We have checked the simulated land sea thermal gradient. Our new analysis shows a weakening land sea gradient at the 850 hPa (e.g. Sinha et al., 2015; Roxy et al., 2015) during LIA compared to MWP in five out of eight models, and also in the upper troposphere (e.g. Goswami et al., 2006; Wang et al., 2013). . Having said that, it is difficult to say whether this is related to the decadal circulation changes associated with ENSO, or independent of them. We cannot also comment whether such changes are associated with external forcings such as volcanoes, unless we conduct sensitivity experiments with AGCMs. Unfortunately, carrying out such experiments is beyond the scope of the current study. These aspects are also reflected in the revised text.

Proxy data analysis (and model experiments) by Schurer et al., (2012) suggest that, the solar changes and increased volcanism are relevant for the climate conditions since 1400 CE. The greenhouse gases may have played a role even around 1600AD and later. However, only half of the proxy datasets used by them suggest any such role of external forcings during the medieval warm period. The paper by Schurer et al., (2012) also suggests that models are unable to reproduce the warming associated with such external forcing around 1000CE. Phipps et al (2013) suggest that detectable weak volcanic signal in Northern Hemisphere temperatures during last 1500 years, but a strong and robust volcanic signal in Southern Hemisphere (Phipps et al., 2013). They claim, bases on their model-cum-proxy data analysis that greenhouse gases, solar irradiance, and volcanic eruptions all influence the mean state of the central Pacific, but there is no evidence that natural or anthropogenic forcings have any systematic impact on ENSO. Just for information, Eurasian snow cover, land sea contrast, Atlantic variability, etc. are some of the other forcings suggested in addition to the tropical Indo-pacific drivers.

Sinha, A.Âăet al. Trends and oscillations in the Indian summer monsoon rainfall over the last two millennia.ÂăNat. Commun.Âă6:6309 doi: 10.1038/ncomms7309 (2015).

Goswami, B. N., Madhusoodanan, M. S., Neema, C. P. & Sengupta, D. A physical mechanism for North Atlantic SST influence on the Indian summer monsoon. Geophys. Res. Lett. 33, L02706 (2006). Wang et al., 2013. Northern Hemisphere summer monsoon intensified by mega-El Nino/southern oscillation and Atlantic Multidecadal oscillation. Proc. Natl. Acad. Sci. USA 110, 5347–5352. Schurer et al., 2012. Separating Forced from Chaotic Climate Variability over the Past Millennium. 6954 JOURNAL OF CL IMATE VOLUME 26. DOI: 10.1175/JCLI-D-12-00826.1 Phipps et al., 2013. Paleoclimate Data–Model Comparison and the Role of Climate Forcings over the Past 1500 Years. 2013 American Meteorological Society. DOI: 10.1175/JCLI-D-12-00108.1